# Fatigue and Related Sleep Disturbances in Hemodialysis Patients: Prevalence, Associated Factors, and the Influence of Nutritional Status

**DOI:** 10.3390/nu18010124

**Published:** 2025-12-30

**Authors:** Ana Casaux-Huertas, María Luz Sánchez-Tocino, Marta San Juan-Miguelsanz, Julia Audije-Gil, Neydu Romero-Lugo, Sonia Muñoz-Pilar, Fabiola Da Pena-Vielba, David Hernan-Gascueña, Paula Manso-Del Real, Soraya Escribano-Loma, Cristina Cubas Sánchez-Beato, María Dolores Arenas-Jiménez

**Affiliations:** 1Department of Nursing, University of Valladolid, 47005 Valladolid, Spain; ana.casaux@uva.es; 2Department of Nursing and Physiotherapy, University of Salamanca, 37008 Salamanca, Spain; 3Institute for Biomedical Research of Salamanca (IBSAL), 37007 Salamanca, Spain; 4Fundación Renal Española, 37500 Salamanca, Spain; 5Fundación Renal Española, 40003 Segovia, Spain; msanjuan.miguelsanz@fundacionrenal.es (M.S.J.-M.); neydu.romero@fundacionrenal.es (N.R.-L.); smunoz@fundacionrenal.es (S.M.-P.); 6Fundación Renal Española, 28003 Madrid, Spain; julia.audije@fundacionrenal.es (J.A.-G.); fdapena@fundacionrenal.es (F.D.P.-V.); dhernan@fundacionrenal.es (D.H.-G.); pmanso@fundacionrenal.es (P.M.-D.R.); mdarenas@fundacionrenal.es (M.D.A.-J.)

**Keywords:** fatigue, sleep disturbances, hemodialysis, chronic kidney disease, nutritional status, MNA-SF

## Abstract

Post-dialysis fatigue is one of the most frequent and limiting symptoms among patients undergoing hemodialysis (HD), characterized by intense physical exhaustion that may persist beyond the treatment session. Sleep disturbances frequently coexist with fatigue and may contribute to overall symptom burden. Nutritional status has been identified as a potential determinant of fatigue severity. Understanding these relationships may help identify associated factors and guide multidisciplinary interventions. **Objectives:** To assess the prevalence and intensity of fatigue in patients receiving HD, to describe the burden of sleep disturbances, and to analyze their association with nutritional status and various clinical, dialytic, and sociodemographic variables. **Methods:** A cross-sectional descriptive study was conducted between November and December 2024 in adults with chronic kidney disease undergoing maintenance HD. Fatigue and sleep disturbances were assessed using brief patient-reported outcome items adapted from PROMIS item bank concepts and analyzed as separate subscales. Nutritional status was evaluated using the Mini Nutritional Assessment–Short Form (MNA-SF). Sociodemographic, clinical, dialytic, and laboratory variables were collected. Statistical analyses were performed using SPSS v29, applying association and correlation tests (*p* ≤ 0.05). **Results:** A total of 729 patients were included (67.1% men), with a mean age of 67.7 ± 14.5 years. Clinically relevant fatigue was reported by approximately 50% of participants, with around 20% presenting severe fatigue. Sleep disturbances affected nearly 60% of patients, with severe impairment reported in approximately 30%. Regarding nutritional status, 61.9% had normal nutrition, 33.2% were at risk of malnutrition, and 4.9% were malnourished. Fatigue was significantly associated with female sex (*p* < 0.001), longer time on hemodialysis (*p* < 0.001), greater weekly dialysis exposure (*p* = 0.012), and poorer nutritional status (*p* = 0.003). The absence of residual urine output showed a borderline association with fatigue (*p* = 0.059) but was significantly associated with sleep disturbances (*p* = 0.002). Sleep disturbance scores were also significantly associated with lower levels of albumin, total proteins, and transferrin. No associations were observed between fatigue and age, BMI, comorbidity, ultrafiltration rate, or biochemical parameters. **Conclusions:** Fatigue is a highly prevalent and clinically relevant symptom in patients undergoing HD and is closely associated with nutritional status and dialysis-related factors. Sleep disturbances are also highly prevalent and may act as an important modulating factor, potentially amplifying fatigue, particularly in patients with greater biological vulnerability or loss of residual kidney function. The systematic use of patient-reported outcome measures (PROMs) to assess fatigue and sleep, together with nutritional evaluation, may facilitate the early identification of vulnerable patients and guide targeted strategies to reduce symptom burden and improve quality of life.

## 1. Introduction

Post-dialysis fatigue is one of the most prevalent and limiting symptoms among individuals with chronic kidney disease (CKD) undergoing hemodialysis (HD). It is characterized by intense exhaustion that appears after the dialysis session, may persist for hours, and does not improve with rest, thereby significantly impairing functional capacity, treatment adherence, and quality of life. The literature consistently reports a prevalence between 60% and 80%, with a substantial proportion of patients experiencing severe fatigue [1,2,3].

The etiology of fatigue in HD is complex and multifactorial. Chronic inflammation, anemia, fluid overload or depletion, hemodynamic instability during treatment, comorbidities, frailty, psychosocial factors, and demographic variables such as female sex have all been implicated, although findings remain heterogeneous and sometimes contradictory across studies [3,4].

Fatigue is also closely related to sleep quality. Patients on HD frequently experience sleep disturbances, which exacerbate physical and mental exhaustion and amplify the overall symptom burden. Although fatigue and sleep disturbances are strongly interrelated and may share underlying pathophysiological mechanisms, they represent distinct clinical domains, highlighting the need to address fatigue within a biopsychosocial framework [5,6].

In recent years, nutritional status has gained central relevance in the understanding of fatigue in HD. Protein–energy wasting and sarcopenia are highly prevalent in this population, affecting a substantial proportion of patients undergoing hemodialysis, with reported prevalence estimates ranging from approximately 30% to over 50% depending on the diagnostic criteria used, and are associated with reduced functional capacity, decreased muscle mass, increased systemic inflammation, and poorer quality of life, all of which contribute to greater fatigue severity [7,8,9]. Cachexia and protein–energy wasting (PEW), as conceptualized within the malnutrition–inflammation–cachexia syndrome (MICS) model, represent key mechanisms linking malnutrition, inflammation, muscle loss, and fatigue in advanced CKD and HD [10,11].

Early identification of nutritional risk through validated screening tools such as the Mini Nutritional Assessment–Short Form (MNA-SF) facilitates the detection of vulnerable profiles and supports timely nutritional interventions [12]. However, important gaps remain in understanding the relationship between fatigue and nutritional status, partly due to heterogeneity in measurement tools, limited sample sizes, and inconsistent evidence regarding the role of residual kidney function, dialysis vintage, session duration, and biochemical markers [13].

In this context, the present study aims to assess the prevalence and intensity of fatigue in patients on HD and to analyze its association with nutritional status, as well as with key clinical, dialytic, and sociodemographic variables, while considering the complementary role of sleep disturbances as a related symptom domain within the overall symptom burden. The ultimate goal is to identify clinically relevant patterns that may guide individualized strategies to improve the overall well-being of this population.

## 2. Materials and Methods

### 2.1. Study Design

A cross-sectional descriptive study was conducted between November and December 2024 in patients with chronic kidney disease (CKD) undergoing maintenance hemodialysis (HD).

### 2.2. Population and Sample

Patients ≥ 18 years old receiving HD treatment at centers of the Fundación Renal Española (FRE) were eligible. Participants were required to provide informed consent and be able to complete the questionnaires. Individuals with severe cognitive impairment or limitations preventing adequate questionnaire completion were excluded. A convenience sampling strategy was used, including all patients who met the eligibility criteria during the study period.

### 2.3. Variables

The following variables were collected:**Sociodemographic variables:** sex and age.**Clinical variables:** Clinical variables included etiology of CKD, body mass index (BMI), Charlson Comorbidity Index (CCI), and time on HD (months). Residual kidney function was defined as the remaining renal capacity to maintain urine production and to contribute to the elimination of solutes and excess fluid in patients undergoing hemodialysis. Classically, residual kidney function refers to the preservation of renal function associated with ongoing urine production (commonly defined as >100 mL/day) after dialysis initiation, regardless of the highly variable amount of uremic toxins excreted in urine at that stage [14]. In the present study, residual urine output was assessed based on patient-reported ongoing urine production at the time of evaluation (yes/no).**Dialysis-related parameters:** vascular access type, session duration (hours), number of weekly sessions, interdialytic weight gain, prescribed ultrafiltration (UF), UF > 10 mL/kg/h [15], delivered Kt per session, and achievement of target Kt (men > 45 L; women > 40 L) [16].**Analytical parameters:** hemoglobin, total cholesterol (HDL and LDL), albumin, total proteins, transferrin, transferrin saturation index, and 25-OH vitamin D.**Primary study variables****Fatigue****Sleep disturbances****Nutritional status**

### 2.4. Data Collection and Measurement Instruments

Fatigue and sleep disturbances were assessed using a **Patient-Reported Outcome Measures (PROMs)** questionnaire composed of items adapted from concepts within the PROMIS^®^ item bank [17].

A total of seven items were used. The fatigue subscale comprised four items focused on post-hemodialysis fatigue and functional impact, specifically assessing: (1) tiredness after the hemodialysis session, (2) the need to lie down or rest after the session, (3) limitations in performing basic activities following dialysis, and (4) the effort required to maintain a conversation. The sleep disturbances subscale included three items referring to the previous seven days, evaluating: (1) poor sleep quality, (2) difficulty falling asleep, and (3) early morning awakening.

These items do not correspond to any of the officially validated PROMIS short forms, and no PROMIS T-scores were calculated; therefore, results were analyzed using summed or mean scores of brief subscales.

Fatigue was considered the primary outcome variable of the study, given its central clinical relevance as one of the most frequent and limiting symptoms in patients undergoing hemodialysis [1,2,3,4]. Sleep disturbances were analyzed as a related domain, considering their close association with fatigue perception and overall symptom burden in this population [5,6].

Both domains were analyzed separately, in order to avoid attributing psychometric validity to a non-validated composite index and to independently explore their associations with the clinical, dialytic, and nutritional variables evaluated.

Nutritional status was assessed using the **Mini Nutritional Assessment (MNA)**, a validated and widely used tool for detecting risk of malnutrition, particularly in adult and older populations with chronic disease [18]. To reduce respondent burden, the six-item version **(Mini Nutritional Assessment–Short Form, MNA-SF)** was used, classifying patients as having normal nutritional status (12–14 points), at risk of malnutrition (8–11 points), or malnutrition (0–7 points) [18,19].

The MNA-SF evaluates key domains related to nutritional status, including recent food intake and appetite, unintentional weight loss, mobility, presence of acute disease or psychological stress, neuropsychological problems, and body mass index or calf circumference. These components allow the identification of nutritional risk by capturing both dietary intake and functional aspects related to nutrition, which are particularly relevant in patients with chronic disease.

The MNA-SF was used as a nutritional screening tool, and not as a diagnostic instrument for protein–energy wasting.

### 2.5. Statistical Analysis

Normality was assessed using the Lilliefors (Kolmogorov–Smirnov) test. Quantitative variables with normal distribution were expressed as mean ± standard deviation (SD) and compared using Student’s *t*-test (two groups) or analysis of variance (ANOVA) for more than two groups. Non-normal variables were expressed as median and interquartile range (P25–P75) and analyzed with Mann–Whitney U or Kruskal–Wallis tests, as appropriate.

Associations between quantitative variables were examined using Pearson’s correlation coefficients (r) for normally distributed data or Spearman’s coefficients (ρ) otherwise. Qualitative variables were expressed as frequencies and percentages and compared using chi-square (χ^2^) tests.

Statistical analyses were performed considering the fatigue and sleep disturbance scores as independent symptom domains, derived from the PROMIS^®^ system. In both cases, associations with sociodemographic, clinical, dialytic, and analytical variables included in the study were explored using the appropriate statistical tests according to variable type and data distribution.

Internal consistency of the ad hoc subscales was assessed to support item aggregation. Cronbach’s alpha coefficients were calculated for the fatigue subscale (4 items) and the sleep disturbances subscale (3 items). Both subscales demonstrated acceptable internal consistency (Cronbach’s α = 0.80; *N* = 696; Cronbach’s α = 0.78; *N* = 696).

For the analysis of the relationship between fatigue and sleep disturbance scores and nutritional status categories defined by the MNA-SF, the Kruskal–Wallis test for independent samples was used. When statistically significant global differences were identified, post hoc pairwise comparisons were performed applying Bonferroni correction for multiple testing. To evaluate whether fatigue and sleep disturbance scores exhibited an ordered trend across ordered categories of nutritional status (normal nutritional status, at risk of malnutrition, and malnutrition), the Jonckheere–Terpstra test was applied.

Statistical significance was set at *p* < 0.05. Analyses were performed using IBM SPSS Statistics version 29.0.1.0 and R with the R Commander user interface (version 4.4.2).

### 2.6. Ethical Considerations

The study was approved by the FRE institutional authorities and the corresponding Ethics Committee. Data confidentiality and compliance with Spanish Law 3/2018 on Personal Data Protection and digital rights were ensured. All participants provided written informed consent and retained their ARCO rights (access, rectification, cancelation, and objection).

## 3. Results

### 3.1. Sample Description

The sample consisted of 729 patients, with a mean age of 67.7 ± 14.5 years, and a predominance of men (489; 67.1%). The most frequent etiology of chronic kidney disease was diabetes mellitus (188; 26.4%), followed by unknown causes and vascular disease. Patients had been on hemodialysis for a median of 57 months (P25–P75: 16–67.5), indicating a cohort with long-standing exposure to renal replacement therapy. The main laboratory parameters showed stable values: hemoglobin 11.4 ± 1.3 g/dL, albumin 4.0 ± 0.4 g/dL, total cholesterol 139.8 ± 39.6 mg/dL, and 25-OH vitamin D 567 ± 434 ng/mL. The remaining sociodemographic, clinical, analytical, nutritional and fatigue-related variables are presented in Table 1.

### 3.2. Results of the Main Variables: Fatigue, Sleep Disturbances, and Nutritional Status

The distribution of fatigue scores showed that a substantial proportion of patients experienced moderate to high levels of fatigue following the hemodialysis session (Figure 1A). At the item level (Figure 1B), the highest mean scores were observed for items related to post-dialysis physical tiredness and feelings of exhaustion during the previous week, whereas the lowest values corresponded to items assessing the effort required to maintain a conversation or engage in social activities. This pattern suggests that fatigue in this cohort manifests predominantly as post-dialysis physical exhaustion, with a comparatively lower impact on social or communicative domains.

Regarding sleep disturbances, their distribution revealed a notable frequency of symptoms related to impaired nocturnal rest and difficulty initiating or maintaining sleep (Figure 1A). Item-level analysis (Figure 1B) showed higher mean scores for items associated with non-restorative sleep and insufficient rest, indicating that sleep disturbances represent a relevant component of symptom burden in patients undergoing hemodialysis and may contribute to the overall perception of fatigue.

With respect to nutritional status, the distribution according to the Mini Nutritional Assessment–Short Form (MNA-SF) is shown in Figure 1C. A total of 431 patients (61.9%) demonstrated normal nutritional status, 231 patients (33.2%) were classified as at risk of malnutrition, and 34 patients (4.9%) met criteria for malnutrition. Item-level analysis (Figure 1D) revealed higher scores in domains related to unintentional weight loss and reduced appetite or digestive problems, whereas lower scores were observed for items referring to psychological stress or recent acute illness and calf circumference. Overall, these findings indicate that nutritional impairment in this cohort manifests primarily through changes in appetite, digestive symptoms, and weight-related alterations, rather than through functional or psychosocial dimensions.

### 3.3. Relationship Between the Main Variables: Fatigue, Sleep Disturbances, and Nutritional Status

Fatigue scores showed statistically significant differences across nutritional status groups defined by the MNA-SF (*p* = 0.003) (Table 2). Patients with normal nutritional status exhibited the lowest levels of fatigue (median 8 [5.5–11]), which increased in the group at risk of malnutrition (9 [6,7,8,9,10,11,12]) and reached the highest values in patients with malnutrition (11 [7,8,9,10,11,12,13]). This pattern reflects a clear clinical gradient, whereby worsening nutritional status is associated with greater fatigue severity.

Similarly, sleep disturbance scores also differed significantly across nutritional status groups (*p* = 0.003) (Table 2). Patients with normal nutritional status showed lower levels of sleep disturbances (median 7 [5,6,7,8,9]), whereas scores increased among patients at risk of malnutrition (8 [5,6,7,8,9,10,11]) and were highest in the malnutrition group (9.5 [6,7,8,9,10,11]). These findings indicate that nutritional deterioration is associated not only with increased fatigue but also with greater impairment of sleep.

Overall, the results indicate that both fatigue and sleep disturbances increase progressively as nutritional status worsens, with consistent differences observed in mean, median, and score distributions across the nutritional status categories.

Post hoc analysis of fatigue scores according to nutritional status (Figure 2A) showed statistically significant differences between patients with normal nutritional status and those with malnutrition (Bonferroni-adjusted *p* = 0.017), while the difference between the normal nutritional status group and the at-risk of malnutrition group approached statistical significance (Bonferroni-adjusted *p* = 0.058). No significant differences were observed between the at-risk of malnutrition and malnutrition groups (Bonferroni-adjusted *p* = 0.307). Regarding sleep disturbance scores (Figure 2B), post hoc analysis revealed significant differences between patients with normal nutritional status and those at risk of malnutrition (Bonferroni-adjusted *p* = 0.006), with no statistically significant differences observed in the remaining pairwise comparisons (Bonferroni-adjusted *p* > 0.05).

In addition, a significant increasing trend in fatigue scores was observed across worsening nutritional status categories (Jonckheere–Terpstra test; *p* < 0.001). Similarly, sleep disturbance scores demonstrated a significant increasing trend with poorer nutritional status (*p* < 0.001). Taken together, these findings indicate that both fatigue and sleep disturbances are associated with nutritional status, although the pattern of differences across nutritional categories differs between the two domains.

### 3.4. Relationship Between Fatigue, Sleep Disturbances, and the Remaining Clinical, Dialytic, and Analytical Variables

Regarding sociodemographic variables (Table 3), no significant associations were observed between age and either fatigue or sleep disturbance scores. However, significant differences according to sex were identified, with women showing higher scores for both fatigue (*p* < 0.001) and sleep disturbances (*p* = 0.039) compared with men.

Among clinical variables, both fatigue and sleep disturbance scores were significantly associated with time on hemodialysis, showing positive correlations in both domains (*p* < 0.001 in both cases). In addition, the absence of residual urine output was significantly associated with higher sleep disturbance scores (*p* = 0.002), whereas the association with fatigue showed a trend toward statistical significance (*p* = 0.059). No significant associations were observed with body mass index or the Charlson Comorbidity Index.

With respect to dialytic parameters, weak but statistically significant associations were observed between fatigue scores and session duration (*p* = 0.049) as well as weekly dialysis minutes (*p* = 0.012). In contrast, sleep disturbance scores were not significantly associated with the dialytic parameters evaluated. No relevant associations were found with delivered Kt, achievement of the Kt target, interdialytic weight gain, or ultrafiltration rate.

Fatigue scores were not significantly associated with most analytical parameters, except for transferrin, which showed a weak inverse correlation. In contrast, sleep disturbance scores were significantly associated with lower albumin (*p* = 0.005), total protein (*p* = 0.049), and transferrin (*p* = 0.016) levels. No significant associations were identified with hemoglobin, lipid profile, transferrin saturation, or 25-OH vitamin D levels.

## 4. Discussion

The present study evaluated the prevalence and intensity of fatigue in patients on hemodialysis (HD), as well as its relationship with nutritional status and clinical and sociodemographic variables, while also examining sleep disturbances as a related symptom domain. The results show that fatigue—conceived as the subjective perception of physical or mental tiredness that affects functioning and well-being—is a highly prevalent and clinically relevant symptom in this population. Higher fatigue intensity was consistently associated with poorer nutritional status, longer cumulative time on HD, and greater weekly dialysis time. Although the absence of residual urine output showed only a borderline association with fatigue, it was significantly associated with sleep disturbances, suggesting that impaired sleep may represent an important pathway through which the loss of residual kidney function contributes to overall symptom burden. In contrast, biochemical parameters and traditional clinical variables did not show significant relationships with fatigue. Taken together, these findings reinforce the multifactorial nature of fatigue and the need to interpret it as an integral indicator of the overall condition of patients on HD.

In our cohort, approximately half of the patients presented clinically relevant fatigue, and one out of five reported severe fatigue. These estimates are consistent with previous studies that identify fatigue as one of the most persistent and disabling symptoms in advanced kidney disease, with prevalence rates ranging from 60% to 80% in different hemodialysis populations [1,2,4,17].

In parallel, sleep disturbances were also highly prevalent, affecting nearly 60% of patients, with approximately 30% reporting severe sleep impairment. Although fatigue was the primary outcome of the study, the high burden of sleep disturbances observed suggests a potential amplifying role of impaired sleep on fatigue perception, reinforcing the close interplay between these two symptom domains in HD [5,6].

The symptom profile revealed a predominance of post-dialysis physical exhaustion and alterations in nocturnal rest, with less impact on social functioning. This pattern is consistent with previous studies describing fatigue in HD mainly as physical exhaustion closely linked to sleep quality rather than as a predominantly cognitive or social phenomenon [5,6,18]. Together, these findings underscore the multidimensional nature of fatigue and highlight the relevance of both physical and sleep-related domains in its clinical expression [6,19].

Although most patients presented an apparently normal nutritional status according to the MNA-SF, one-third were at risk of malnutrition and a smaller proportion met criteria for established malnutrition, in line with previous studies [20]. In the present analysis, fatigue scores increased progressively as nutritional status worsened, replicating findings that link malnutrition, loss of muscle mass, and protein–energy wasting with greater fatigue intensity [8,9,21]. These results can be explained by shared mechanisms—lower energy reserves, reduced lean mass, chronic inflammation, and poorer functional capacity—all of which are involved in advanced CKD [22].

In addition, sleep disturbances also differed across nutritional status categories, suggesting that nutritional impairment may contribute not only directly to fatigue but also indirectly through its impact on sleep quality. Previous studies in patients undergoing hemodialysis have reported that poorer nutritional status, as assessed by malnutrition–inflammation–based indices and related nutritional markers, is associated with worse sleep quality and a higher burden of sleep disturbances [23]. These findings support the hypothesis that nutritional deterioration may exacerbate fatigue both through direct metabolic and functional pathways and indirectly by impairing sleep, thereby amplifying overall symptom burden in this population [24].

With regard to clinical variables, in our cohort women presented significantly higher levels of fatigue, a finding consistent with previous evidence [4,13]. Women also showed significantly higher levels of sleep disturbances, reinforcing the notion of a sex-related vulnerability to symptom burden in HD. This difference has been attributed to the interaction of biological, hormonal, and body composition factors, as well as gender roles and patterns of emotional expression that may favor greater recognition and reporting of symptoms by women.

With regard to age, no significant differences were found between patients older and younger than 65 years. The literature shows heterogeneous results, with studies reporting higher fatigue in older people, others in younger individuals, and others with no clear association; in our analysis, no significant differences were found in post-dialysis fatigue [25]. Taken together, these results suggest that chronological age, by itself, may not be a main determinant of fatigue and that other factors such as functional status, frailty, or psychosocial situation may carry greater weight.

Cumulative time on HD showed a moderate association with fatigue intensity, consistent with studies describing a higher symptom burden in patients with more years on treatment, related to the progression of comorbidities, psychological strain, and loss of muscle mass [26,27]. However, its isolated effect was limited, indicating that although dialysis vintage may contribute to increased fatigue, it does not fully explain it and should be interpreted in the context of other clinical and nutritional determinants.

Importantly, no significant associations were observed between fatigue and classical biochemical parameters such as albumin, hemoglobin, or other analytical markers, despite the clear relationship between fatigue and nutritional status. This apparent discrepancy may be explained by several factors: the cross-sectional design of the study, the relative clinical stability of the cohort, and the well-recognized limitations of traditional biochemical markers, which often act as late or indirect indicators of nutritional and inflammatory status [13]. In contrast, sleep disturbances were significantly associated with lower levels of albumin, total proteins, and transferrin, suggesting that biochemical alterations may be more closely related to sleep impairment than to fatigue itself, and that sleep disturbances could represent a more sensitive intermediary domain linking biological vulnerability and subjective fatigue perception [28,29].

One of the clinically relevant findings of the present study was the greater symptom burden observed in patients without residual urine output. Although the association between the absence of residual diuresis and fatigue showed only a borderline effect, the loss of residual kidney function was significantly associated with greater sleep disturbances, suggesting a potential indirect impact on fatigue perception.

Previous evidence has consistently shown that residual kidney function is associated with better quality of life, lower symptom burden, reduced systemic inflammation, improved volume and electrolyte control, and lower cardiovascular risk [30,31,32]. From a pathophysiological perspective, the loss of residual filtration may promote the retention of uremic solutes, greater interdialytic fluid overload, and the occurrence of nocturnal symptoms such as dyspnea, cramps, or restlessness, all of which can disrupt sleep [33]. Impaired nocturnal rest may, in turn, amplify daytime tiredness and limit recovery after hemodialysis sessions, thereby contributing to persistent fatigue [34].

In this context, our findings suggest that residual kidney function may exert a protective effect on symptomatic well-being not only through direct mechanisms, but also indirectly through its impact on sleep quality. These results underscore the clinical importance of strategies aimed at preserving residual kidney function, not only from a survival perspective but also to improve symptom control and patient-reported outcomes in hemodialysis [35].

Regarding dialytic variables, longer dialysis session duration and greater weekly dialysis time were significantly associated with higher fatigue levels. This finding can be interpreted in terms of “treatment burden,” since more time connected to the machine implies prolonged immobility, restrictions, and intradialytic symptoms, factors previously described as contributors to exhaustion [36,37,38]. Even so, its impact was modest, suggesting that its influence is secondary compared with major determinants such as nutrition and residual kidney function.

In contrast, high ultrafiltration (>10 mL/kg/h) did not show a significant association with fatigue, despite its known relationship with hemodynamic instability, adverse cardiovascular events, and higher mortality [15]. This result partially contrasts with studies that have described the relationship between high ultrafiltration rates, intradialytic hypotension, cramps, and physical discomfort [3,18,39]. This finding suggests that mechanisms affecting cardiovascular prognosis do not always translate into greater perception of tiredness, possibly due to progressive adjustment of dry weight and individualized management of ultrafiltration.

Regarding dialysis dose, no clear relationship was observed between Kt, considered as a continuous variable, and fatigue levels, in line with studies that have not demonstrated a consistent association between clearance indices and symptoms [25,40,41]. In our analysis, achieving the target Kt was associated with slightly higher levels of fatigue, a finding that, although difficult to interpret, could be related to the hypothesis that more intensive clearance also entails greater losses of amino acids and other energy substrates, favoring the onset of fatigue in patients who are already vulnerable from a nutritional standpoint, as has been proposed in studies analyzing amino acid homeostasis and the sensation of tiredness in HD [20]. Nevertheless, the cross-sectional and exploratory nature of our study does not allow causal relationships to be established, and these results should be interpreted with caution.

Finally, no significant associations were identified between fatigue and variables such as BMI, Charlson Comorbidity Index, hemoglobin, albumin, or other analytical parameters. These findings are consistent with recent reviews that highlight the limited ability of isolated markers to explain fatigue [13], probably due to its multifactorial nature and the simultaneous influence of biological, functional, psychological, and social factors [3,42].

Taken together, the results support the interpretation of fatigue as a complex symptom in which nutritional, functional, emotional, and dialysis treatment–related determinants converge, rather than as a phenomenon directly attributable to a single clinical parameter. Its high prevalence and its association with modifiable factors reinforce its usefulness as a sensitive marker of the overall status of patients on HD.

The findings underscore the need to routinely integrate fatigue assessment with nutritional evaluation and monitoring of residual diuresis, in order to identify profiles of greater vulnerability and guide personalized interventions aimed at optimizing nutritional support, adapting dialysis parameters, and promoting strategies that reduce symptom burden.

Looking ahead to future research, it is necessary to delve into the pathophysiological mechanisms linking nutritional deterioration, loss of muscle mass, and fatigue, as well as the role of inflammation, frailty, and sleep disturbances as potential mediators. Longitudinal studies would help clarify the temporal evolution of fatigue and evaluate the impact of specific interventions—such as nutritional optimization, individualized adjustment of dialysis time, or preservation of residual kidney function—on its trajectory. It will also be essential to validate and compare sensitive and specific measurement instruments to detect clinically relevant changes in fatigue in hemodialysis.

This study has several limitations that should be considered when interpreting the results. First, the cross-sectional design precludes establishing causal relationships between fatigue, sleep disturbances, and the clinical, nutritional, or dialytic factors analyzed. Second, some variables were underrepresented, such as high ultrafiltration rates or established malnutrition, which may have reduced the statistical power to detect certain associations.

In addition, potentially relevant variables such as psychological factors (anxiety, depression, or coping strategies), known modulators of both fatigue and sleep disturbances in patients undergoing hemodialysis, were not included. Likewise, multivariable analyses adjusted for specific electrolyte disturbances, which could act as confounding factors, were not performed.

Finally, nutritional status was assessed using the MNA-SF as a screening tool, without incorporating dialysis-specific malnutrition scores such as protein–energy wasting criteria or the Malnutrition–Inflammation Score. Although these instruments provide a more detailed characterization of nutritional status in hemodialysis patients, their use was not feasible in the present study due to the multicenter design and the lack of homogeneous availability of the required variables across participating centers.

Lastly, the heterogeneity of the hemodialysis population and inter-center variability may have influenced some of the observed differences.

## 5. Conclusions

Fatigue is a highly prevalent and clinically relevant symptom in patients undergoing hemodialysis. Its close relationship with nutritional status and dialysis-related factors, such as longer dialysis exposure, highlights the need for a comprehensive clinical approach that simultaneously considers these dimensions.

The present findings also underline the relevance of sleep disturbances, which were highly prevalent and showed significant associations with nutritional and biological markers, as well as with the absence of residual kidney function. While the association between fatigue and residual diuresis was only borderline, impaired sleep may act as an important modulating factor linking the loss of residual kidney function to overall symptom burden and potentially amplifying fatigue.

In this context, the systematic use of patient-reported outcome measures (PROMs) to assess fatigue and sleep disturbances, together with nutritional evaluation and monitoring of residual kidney function, may facilitate the early identification of patients with greater clinical vulnerability. Fatigue emerges as a sensitive marker of the patient’s global condition and a useful target to guide individualized interventions aimed at optimizing treatment parameters, reducing symptom burden, and improving quality of life in patients on hemodialysis.

## Figures and Tables

**Figure 1 nutrients-18-00124-f001:**
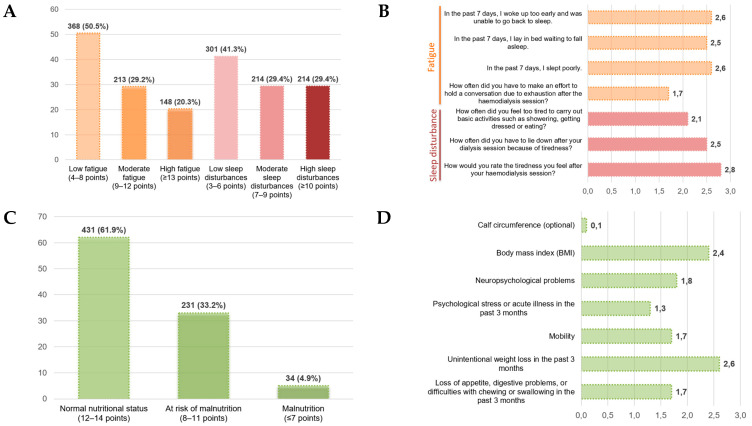
Distribution and item-level profiles of fatigue, sleep disturbances, and nutritional status (MNA-SF). (**A**) Distribution of fatigue and sleep disturbance severity in descriptive categories. (**B**) Item-level mean scores for the fatigue and sleep disturbances. (**C**) Distribution of nutritional status according to the Mini Nutritional Assessment–Short Form (MNA-SF). (**D**) Item-level mean scores of the MNA-SF.

**Figure 2 nutrients-18-00124-f002:**
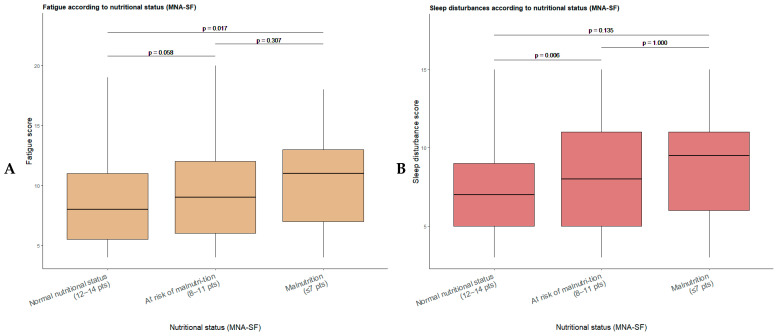
Boxplots of fatigue (**A**) and sleep disturbance (**B**) scores according to nutritional status defined by the Mini Nutritional Assessment–Short Form (MNA-SF). Post hoc pairwise comparisons between nutritional status groups were performed applying Bonferroni correction for multiple testing.

**Table 1 nutrients-18-00124-t001:** Baseline characteristics of the study population (*N* = 729).

Variable	Value
**Sociodemographic variables**	
Age (years): mean ± SD	67.7 ± 14.5
≤65 years, n (%)	244 (33.5%)
≥65 years, n (%)	485 (66.5%)
Sex: male, n (%)	489 (67.1%)
female, n (%)	240 (32.9%)
Nationality: Spanish, n (%)	614 (84.5%)
Migrant, n (%)	113 (15.5%)
**Clinical variables**	
**Etiology of CKD**	
- Diabetes mellitus, n (%)	188 (26.4%)
- Unknown, n (%)	141 (19.8%)
- Vascular, n (%)	111 (15.6%)
- Nephritis/pyelonephritis, n (%)	55 (7.7%)
- Glomerulonephritis/membranous nephropathy, n (%)	113 (15.8%)
- Polycystic kidney disease, n (%)	57 (8.0%)
- Tumor/other causes, n (%)	48 (6.7%)
Body mass index (BMI), mean ± SD	26.5 ± 5.3
Charlson Comorbidity Index (CCI), mean ± SD	7.8 ± 2.9
Time on hemodialysis (months), median (P25–P75)	57.0 (16–67.5)
Residual urine output present, n (%)	140 (44.3%)
Not present	176 (55.7)
**Dialytic parameters**	
**Vascular access in use**	
- Native/prosthetic AVF, n (%)	409 (64.6%)
- Central venous catheter (CVC), n (%)	224 (35.4%)
Session duration (hours), mean ± SD	3.7 ± 0.4
≥4 h sessions, n (%)	155 (21.4%)
Interdialytic weight gain (kg), mean ± SD	2.2 ± 1.3
Ultrafiltration rate (mL/kg/h), mean ± SD	7.9 ± 3.4
Ultrafiltration >10 mL/kg/h, n (%)	185 (26%)
Sessions per week, mean ± SD	2.9 ± 0.4
Kt (L), mean ± SD	51.8 ± 9.2
Achieved Kt target, n (%)	611 (84.2%)
**Laboratory parameters**	
Hemoglobin (g/dL), mean ± SD	11.4 ± 1.3
Total cholesterol (mg/dL), mean ± SD	139.8 ± 39.6
HDL cholesterol (mg/dL), mean ± SD	45.5 ± 14.8
LDL cholesterol (mg/dL), mean ± SD	69.1 ± 31.0
Albumin (g/dL), mean ± SD	4.0 ± 0.4
Total proteins (g/dL), mean ± SD	6.5 ± 0.6
Transferrin (mg/dL), mean ± SD	69 ± 49.4
Transferrin saturation (%), mean ± SD	28 ± 12.3
25-OH vitamin D (ng/mL), mean ± SD	567 ± 434

**Table 2 nutrients-18-00124-t002:** Total fatigue and sleep disturbances score according to nutritional status (MNA-SF).

	Fatigue				Sleep Disturbances			
Nutritional Status (MNA-SF)	Mean ± SD	Median (IQR)	Min–Max	*p* Global *	Mean ± SD	Median (IQR)	Min–Max	*p* Global *
Normal nutritional status (12–14 pts) *(n* = 431)	8.75 ± 3.82	8 (5.5–11)	4–20	**0.003**	7.31 ± 3.25	7 (5–9)	3–15	**0.003**
At risk of malnutrition (8–11 pts) *(n* = 231)	9.51 ± 4.03	9 (6–12)	4–20		8.25 ± 3.65	8 (5–11)	3–15	
Malnutrition (≤7 pts) *(n* = 34)	10.71 ± 4.10	11 (7–13)	4–18		8.47 ± 3.47	9.5 (6–11)	3–15	

* Global *p*-value calculated using the Kruskal–Wallis test for independent samples. Significant associations (*p* < 0.05) are shown in bold.

**Table 3 nutrients-18-00124-t003:** Relationship between fatigue and sleep disturbance scores and sociodemographic, clinical, dialytic, and analytical variables.

Variable/Group	Statistic	*N*	Fatigue Subscale (Mean ± SD/ρ)	*p*-Value	Sleep Disturbance Subscale (Mean ± SD/ρ)	*p*-Value
**Sociodemographic variables**						
Age (years)	Spearman ρ	729	ρ = −0.064	0.086	ρ = −0.001	0.972
Sex: men	Mann–Whitney U	489	**8.0 (6.0–11.0)**	**<0.001**	**7.0 (5.0–10.0)**	**0.039**
women		240	**10.0 (6.8–13.0)**		**8.0 (5.0–11.0)**	
**Clinical variables**						
Body mass index (BMI)	Spearman ρ	719	ρ = 0.006	0.880	ρ = 0.054	0.151
Charlson Comorbidity Index	Spearman ρ	723	ρ = 0.003	0.941	ρ = 0.041	0.270
Time on hemodialysis (months)	Spearman ρ	689	**ρ = 0.221**	**<0.001**	**ρ = 0.130**	**<0.001**
Residual urine output: No	Mann–Whitney U	176	9.0 (6.0–12.3)	0.059	**9.0 (6.0–11.0)**	**0.002**
Yes		140	8.0 (5.0–12.0)		**7.0 (4.0–10.0)**	
**Dialytic parameters**						
Session duration (hours)	Spearman ρ	725	**ρ = 0.073**	**0.049**	ρ = 0.047	0.202
Weekly dialysis minutes	Spearman ρ	722	**ρ = 0.093**	**0.012**	ρ = 0.057	0.128
Kt (L)	Spearman ρ	726	ρ = −0.009	0.807	ρ = 0.000	0.992
Achieving Kt target: No	Mann–Whitney U	115	8.0 (5.0–12.0)	0.071	7.0 (4.0–9.0)	0.114
Yes		611	9.0 (6.0–12.0)		8.0 (5.0–10.0)	
Interdialytic weight gain (kg)	Spearman ρ	721	ρ = 0.061	0.101	ρ = 0.035	0.355
Ultrafiltration rate (mL/kg/h)	Spearman ρ	708	ρ = 0.024	0.515	ρ = −0.002	0.963
Ultrafiltration rate <10 mL/kg/h	Mann–Whitney U	523	8.0 (6.0–12.0)	0.172	7.0 (5.0–10.0)	0.398
≥10 mL/kg/h		185	9.0 (6.0–12.0)		8.0 (5.0–11.0)	
**Analytical parameters**						
Hemoglobin (g/dL)	Spearman ρ	683	ρ = 0.004	0.914	ρ = −0.041	0.290
Total cholesterol (mg/dL)	Spearman ρ	680	ρ = 0.033	0.385	ρ = −0.046	0.234
HDL cholesterol (mg/dL)	Spearman ρ	580	ρ = −0.059	0.156	ρ = −0.042	0.313
LDL cholesterol (mg/dL)	Spearman ρ	575	ρ = 0.011	0.798	ρ = −0.072	0.082
Albumin (g/dL)	Spearman ρ	705	ρ = 0.011	0.774	**ρ = −0.107**	**0.005**
Total proteins (g/dL)	Spearman ρ	609	ρ = −0.037	0.359	**ρ = −0.080**	**0.049**
Transferrin (mg/dL)	Spearman ρ	700	**ρ = −0.093**	**0.014**	**ρ = −0.091**	**0.016**
Transferrin saturation (%)	Spearman ρ	705	ρ = −0.009	0.804	ρ = −0.007	0.856
25-OH vitamin D (ng/mL)	Spearman ρ	663	ρ = −0.007	0.855	ρ = −0.072	0.065

Statistical tests used according to variable type: Spearman or Pearson correlations for continuous variables; independent-samples *t*-test for dichotomous variables. Two-tailed *p*-values are presented. Significant associations (*p* < 0.05) are shown in bold.

## Data Availability

The data presented in this study are not publicly available due to patient privacy and data protection restrictions. Data may be made available upon reasonable request to the corresponding author and with authorization from the Fundación Renal Española.

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
