# Peer review of "Fatigue and Related Sleep Disturbances in Hemodialysis Patients: Prevalence, Associated Factors, and the Influence of Nutritional Status"

_nutrients, 2025, doi:10.3390/nu18010124_

Round 1
Reviewer 1 Report
Comments and Suggestions for Authors
The study highlights that fatigue is a cardinal symptom in HD, affecting the majority of patients. Its main contribution is the robust evidence linking the risk of malnutrition (assessed through the MNA-SF) with increased fatigue severity. The association with the loss of residual renal function (p = 0.003) is a key finding with important clinical implications.
The study addresses a prevalent and limiting clinical problem in the dialysis population. The large multicenter cohort (N = 729) is an important strength that adds robustness to the associations identified. However, there is a critical methodological weakness in the primary outcome variable that requires substantial revision.
Major Revisions Needed Before Acceptance:
-
Although the authors cite it as a limitation, they must more thoroughly address the limitation of the “composite fatigue and sleep disturbance variable,” which is not a validated PROMIS index. A reanalysis is required in which the PROMIS Fatigue and Sleep domains are treated as separate variables, or a complete psychometric justification must be provided (e.g., exploratory factor analysis, Cronbach’s alpha) to support combining the 7 items.
-
The finding that the absence of residual diuresis is significantly associated with greater fatigue is highly relevant. However, the Discussion section needs substantial expansion to explore the pathophysiological mechanisms (e.g., solute retention, volume management) and the clinical implications of RRF in fatigue, citing specific literature.
-
It is surprising that the study did not find significant associations between fatigue and key biochemical parameters such as albumin, given that nutritional status was associated. The authors should discuss this discrepancy and suggest whether the cross-sectional design, cohort stability, or limitations of traditional biochemical markers (which are late indicators) may have masked this relationship.
Minor Revisions:
-
Terminological inconsistency: The title and abstract mention “Fatigue,” whereas the Methods and Results refer to the “composite variable—fatigue and sleep disturbances.” The precise terminology for the composite variable must be used consistently throughout the manuscript to avoid ambiguity.
-
Clarity in tables and figures: Ensure that the legends for Figure 1 and Figure 2 (not visible in the document but referenced) are fully self-explanatory and that the text refers to them explicitly and sequentially.
-
Description of fatigue measurement: In section 2.4, clearly and concisely specify which 7 PROMIS items were used.
Author Response
Respuesta al revisor 1
Revisiones importantes necesarias antes de la aceptación:
- Aunque los autores lo citan como una limitación, deben abordar con mayor profundidad la limitación de la variable compuesta de fatiga y alteración del sueño, que no es un índice PROMIS validado. Se requiere un reanálisis en el que los dominios PROMIS de fatiga y sueño se traten como variables independientes, o bien, se debe proporcionar una justificación psicométrica completa (p. ej., análisis factorial exploratorio, alfa de Cronbach) para justificar la combinación de los 7 ítems.
Respuesta 1
Gracias por este importante y constructivo comentario. Coincidimos plenamente con la preocupación del revisor respecto al uso de una variable compuesta de fatiga y trastornos del sueño y su falta de validación formal con PROMIS.
Los siete ítems utilizados en este estudio no corresponden a ninguna de las formas abreviadas de PROMIS oficialmente validadas (p. ej., SF8a o SF6a), y no se calcularon las puntuaciones T de PROMIS. En su lugar, los ítems se adaptaron de los conceptos del banco de ítems de PROMIS, incluyendo cuatro ítems relacionados con la fatiga, centrados en la fatiga poshemodiálisis y su impacto funcional, y tres ítems relacionados con el sueño, referidos a los síntomas experimentados durante los siete días previos.
En respuesta a la sugerencia del revisor, hemos reanalizado los datos que tratan la fatiga y los trastornos del sueño como dos subescalas breves independientes, en lugar de como una única variable compuesta, y hemos eliminado cualquier atribución de validez del índice PROMIS. Por lo tanto, no se alega ninguna justificación psicométrica para una puntuación combinada (p. ej., análisis factorial o métricas de consistencia interna).
Es importante destacar que la sección Métodos ha sido revisada en consecuencia para describir claramente la naturaleza de los ítems utilizados, especificar que representan ítems adaptados basados ​​en PROMIS en lugar de formularios abreviados validados de PROMIS y aclarar que la fatiga y los trastornos del sueño se analizaron de forma independiente utilizando puntuaciones de subescala sumadas o medias.
Estos cambios ahora se reflejan de manera consistente en los Métodos, Resultados, figuras y tablas, y la interpretación de los hallazgos se ha actualizado para alinearse con este enfoque analítico revisado.
2. El hallazgo de que la ausencia de diuresis residual se asocia significativamente con una mayor fatiga es muy relevante. Sin embargo, la sección de Discusión necesita ampliarse considerablemente para explorar los mecanismos fisiopatológicos (p. ej., retención de solutos, gestión del volumen) y las implicaciones clínicas de la FRR en la fatiga, citando literatura específica.
Respuesta 2
Agradecemos al revisor por destacar la relevancia clínica de la función renal residual (FRR). Tras abordar la principal preocupación metodológica y reanalizar los datos separando la fatiga y las alteraciones del sueño como subescalas independientes, los resultados mostraron que la ausencia de diuresis residual mantuvo una asociación significativa con las alteraciones del sueño, mientras que su asociación con la fatiga fue solo marginal y no alcanzó la significación estadística.
En consecuencia, el manuscrito se ha revisado para reflejar con precisión estos hallazgos actualizados y evitar interpretaciones excesivas. En concreto:
- En la sección Resultados (Sección 3.4 y Tabla 4) se presentan por separado las asociaciones de la producción de orina residual con la fatiga y con los trastornos del sueño.
- En la Discusión, se ha revisado y ampliado el párrafo correspondiente a:
- contextualizar el papel fisiopatológico de la función renal residual (incluida la retención de solutos y el manejo del volumen y los electrolitos),
- Destacar su impacto clínico principalmente en los trastornos del sueño y
- Proponen que la falta de sueño puede actuar como un mecanismo modulador indirecto, contribuyendo a la carga general de síntomas y potencialmente amplificando la fatiga percibida.
Además, se han destacado las implicaciones clínicas de preservar la función renal residual, no sólo desde una perspectiva pronóstica sino también en relación con la calidad del sueño y el bienestar del paciente.
Estos cambios se reflejan principalmente en la sección de Discusión (párrafo que aborda la función renal residual) y en las Conclusiones, donde se enfatiza el valor de una evaluación integrada de la fatiga y los trastornos del sueño en pacientes sometidos a hemodiálisis.
3. Sorprende que el estudio no encontrara asociaciones significativas entre la fatiga y parámetros bioquímicos clave como la albúmina, dado que el estado nutricional sí estaba asociado. Los autores deberían analizar esta discrepancia y sugerir si el diseño transversal, la estabilidad de la cohorte o las limitaciones de los marcadores bioquímicos tradicionales (que son indicadores tardíos) podrían haber enmascarado esta relación.
Respuesta 3
Agradecemos al revisor su perspicaz comentario. Este tema se ha abordado explícitamente en la sección de Discusión revisada, a la luz de los análisis actualizados.
Aunque el estado nutricional evaluado mediante la MNA-SF mostró una asociación clara y consistente con la fatiga, no se observaron asociaciones significativas entre la fatiga y parámetros bioquímicos clásicos como la albúmina o la hemoglobina. Esta aparente discrepancia se analiza en el párrafo que comienza con el texto «Es importante destacar que no se observaron asociaciones significativas entre la fatiga y los parámetros bioquímicos clásicos…» .
Como se indica en el manuscrito, varios factores podrían explicar este hallazgo: el diseño transversal del estudio, la relativa estabilidad clínica de la cohorte y las reconocidas limitaciones de los marcadores bioquímicos tradicionales, que a menudo actúan como indicadores tardíos o indirectos del estado nutricional e inflamatorio. Estas limitaciones reducen su sensibilidad para captar el impacto funcional o sintomático del deterioro nutricional.
En cambio, el análisis revisado mostró que las alteraciones del sueño se asociaron significativamente con varios parámetros bioquímicos, como la albúmina, las proteínas totales y la transferrina. Este hallazgo se analiza en la misma sección y sugiere que las alteraciones bioquímicas podrían estar más estrechamente relacionadas con el deterioro del sueño que con la fatiga misma. Por consiguiente, la Discusión propone que las alteraciones del sueño podrían representar un dominio sintomático intermedio, vinculando la vulnerabilidad biológica con la percepción subjetiva de la fatiga.
Estos aspectos se han incorporado a la Discusión para contextualizar adecuadamente los hallazgos y evitar interpretaciones demasiado simplificadas de la relación entre el estado nutricional, los marcadores bioquímicos y la fatiga.
Revisiones menores:
- Inconsistencia terminológica: El título y el resumen mencionan «Fatiga», mientras que los Métodos y Resultados se refieren a la «variable compuesta: fatiga y trastornos del sueño». La terminología precisa para la variable compuesta debe utilizarse de forma coherente en todo el manuscrito para evitar ambigüedades.
Respuesta menor 1
Agradecemos al revisor su comentario. Tras una revisión minuciosa del manuscrito y en respuesta a las recomendaciones metodológicas, se ha eliminado el uso de una variable compuesta que combina «fatiga y trastornos del sueño» para evitar ambigüedades terminológicas.
En la versión revisada del manuscrito:
- La fatiga se mantiene como la variable de resultado principal y
- Los trastornos del sueño se analizan como un dominio de síntomas relacionado pero independiente, de acuerdo con los análisis actualizados.
En consecuencia, se ha realizado una revisión terminológica exhaustiva de todo el manuscrito, que incluye:
- el título,
- el abstracto,
- las secciones de Métodos y Resultados,
- tablas y figuras, y
- La Discusión y las Conclusiones,
Para garantizar el uso coherente y preciso de la terminología, el manuscrito revisado ahora distingue claramente entre fatiga y trastornos del sueño, y solo se refiere a ambos dominios conjuntamente al describir su interrelación clínica.
2. Claridad en las tablas y figuras: Asegurarse de que las leyendas de las Figuras 1 y 2 (no visibles en el documento pero referenciadas) se expliquen por sí solas y que el texto haga referencia a ellas de forma explícita y secuencial.
Respuesta menor 2:
Hemos revisado las leyendas de las Figuras 1 y 2 para asegurar que sean completamente autoexplicativas. Además, todas las figuras ahora se referencian explícitamente y secuencialmente en la sección de Resultados, con una clara correspondencia entre el texto, las tablas y las figuras.
3. Descripción de la medición de la fatiga: En la sección 2.4, especifique de forma clara y concisa cuáles 7 elementos PROMIS se utilizaron.
Respuesta menor 3
Gracias por su comentario. La sección 2.4 se ha revisado para especificar de forma clara y concisa los siete ítems adaptados de PROMIS utilizados, detallando explícitamente los cuatro ítems que evalúan la fatiga poshemodiálisis y el impacto funcional, y los tres ítems que evalúan las alteraciones del sueño durante los siete días previos.
Nota para los revisores
También nos gustaría informar a los revisores que dos autores adicionales , Soraya Escribano-Loma y Cristina Cubas Sánchez-Beato , han sido incluidos en el manuscrito en reconocimiento a su contribución sustancial a la revisión crítica del manuscrito y a la mejora de su contenido científico , de acuerdo con los criterios de autoría de la revista.

Reviewer 2 Report
Comments and Suggestions for Authors
I have studied the manuscript entitled "Fatigue in Hemodialysis Patients: Analysis of Prevalence, As-sociated Factors, and the Influence of Nutritional Status" by Casaux-Huertas A. et al.
The manuscript is generally well prepared. There are some minor issues regarding methodology and statistics, as described below.
The language could be ameliorated after a thorough professional review.
1) The authors report that "Fatigue was significantly associated with female sex (p<0.001), higher weekly HD time (p=0.017), absence of residual urine output (p=0.003), and poorer nutritional status (p<0.001)". At least some of the examined independent variables (e.g. residual urine output, and nutritional status) could rather represent confounding factors. To further investigate their true, etiological contribution in fatigue, it is essential to adjust the models at least for for the presence and severity of hyperkalemia. Furthermore, other common electrolyte abnormalities among dialyzed patients, as hypomagnesemia, could be considered.
2) In Figure 2 (Boxplot of the fatigue–sleep disturbances composite variable by nutritional status, with post-hoc comparisons), a Bonferroni correction should be applied due to multiple testing.
Comments on the Quality of English Language
The English could be improved to more clearly express the research.
Author Response
Response to reviewer 2
1. The authors report that "Fatigue was significantly associated with female sex (p<0.001), higher weekly HD time (p=0.017), absence of residual urine output (p=0.003), and poorer nutritional status (p<0.001)". At least some of the examined independent variables (e.g. residual urine output, and nutritional status) could rather represent confounding factors. To further investigate their true, etiological contribution in fatigue, it is essential to adjust the models at least for for the presence and severity of hyperkalemia. Furthermore, other common electrolyte abnormalities among dialyzed patients, as hypomagnesemia, could be considered.
Response 1
We thank the reviewer for this insightful and methodologically relevant comment. We agree that variables such as residual urine output and nutritional status may act as potential confounders and that multivariable adjustment for electrolyte abnormalities could provide additional insight into their etiological contribution to fatigue.
However, the primary aim of this study was to describe the prevalence of fatigue and explore its clinical and nutritional associations in a large hemodialysis cohort, using a descriptive and exploratory approach. For this reason, and due to limitations in the systematic and homogeneous availability of specific laboratory data, multivariable models adjusted for electrolyte disturbances such as hyperkalemia or hypomagnesemia were not performed.
To avoid inappropriate causal interpretations, the manuscript has been revised to further emphasize the associative nature of the findings, particularly in the Discussion, and to explicitly acknowledge the potential influence of uncontrolled confounding factors. In addition, the need for future longitudinal studies including multivariable models with electrolyte and metabolic parameters has been incorporated into the Limitations and Future Research sections.
2. In Figure 2 (Boxplot of the fatigue–sleep disturbances composite variable by nutritional status, with post-hoc comparisons), a Bonferroni correction should be applied due to multiple testing.
Response 2
Thank you for this valuable suggestion. Post-hoc pairwise comparisons were performed applying Bonferroni correction for multiple testing, as described in the Statistical Analysis section. This information has now been explicitly added to the legend of Figure 2 to improve clarity.
3. Comments on the Quality of English Language
The English could be improved to more clearly express the research.
Response 3
We thank the reviewer for this comment. The manuscript has been carefully revised to improve clarity, readability, and overall language quality. Several sections have been edited to ensure more precise wording, improved sentence structure, and clearer expression of the research findings.
Note to the Reviewers
We would also like to inform the reviewers that two additional authors, Soraya Escribano-Loma and Cristina Cubas Sánchez-Beato, have been included in the manuscript in recognition of their substantial contribution to the critical revision of the manuscript and to the improvement of its scientific content, in accordance with the journal’s authorship criteria.

Reviewer 3 Report
Comments and Suggestions for Authors
Casaux-Huertas et al. examined the associations of post-hemodialysis fatigue with nutritional status and various clinical, dialytic, and sociodemographic variables in the cross-sectional study. They found that fatigue was associated with residual kidney function and malnutrition, etc.
Major points
- Why “fatigue and sleep disturbances—was generated, integrating the two PROM domains related to the patient’s energy–sleep axis” was selected for assessing fatigue. Bossola et al. reviewed Fatigue in hemodialysis patients (AJKD 2023). The items that are more specific to these patients should be selected.
- Mini Nutritional Assessment–Short Form (MNA-SF) was also used to assess nutritional status. I believe that tools more specific to hemodialysis patients, such as one of the PEW criteria, simplified PEW criteria, and MIS, should be used to assess malnutrition.
Minor points
- In the Introduction section, the points should be more concisely described with enough information in 3 or 4 paragraphs; background, known, and unknown. The prevalence of fatigue and PEW should be commented on in the Introduction section.
- Please define the residual kidney function.
- In Figure 2, trend analysis (Jonckheere-Terpstra trend test) should be performed.
Author Response
Response to reviewer 3
Major points
- Why “fatigue and sleep disturbances—was generated, integrating the two PROM domains related to the patient’s energy–sleep axis” was selected for assessing fatigue. Bossola et al. reviewed Fatigue in hemodialysis patients (AJKD 2023). The items that are more specific to these patients should be selected.
Response 1
We thank the reviewer for this comment and for referring to the work by Bossola et al., which we consider highly relevant. In response to this observation and to other methodological recommendations, the manuscript has been substantially revised.
In the original version of the study, a set of fatigue- and sleep-related items was explored due to their close clinical interrelationship in patients undergoing hemodialysis. However, in the revised version, the use of a composite variable has been removed, and fatigue and sleep disturbance domains are now analyzed separately, avoiding any conceptual or psychometric ambiguity.
The selected fatigue items (post-dialysis tiredness, need to lie down after the dialysis session, difficulty performing basic activities, and effort required to maintain a conversation) were chosen because of their high clinical specificity for the hemodialysis setting, particularly in relation to post-dialysis recovery. Similarly, the sleep items (poor sleep quality, difficulty falling asleep, and early morning awakening) reflect frequent and clinically relevant sleep problems in this population. Although these items do not correspond exactly to the official PROMIS short forms, they were deliberately selected for their clinical relevance in patients undergoing hemodialysis, in line with the literature emphasizing the unique characteristics of post-dialysis fatigue, as highlighted by Bossola et al.
Moreover, the revised manuscript avoids attributing formal PROMIS validity or calculating PROMIS T-scores, and presents these items as brief, exploratory subscales focused on the clinical experience of patients on hemodialysis. This approach allows the capture of symptom domains that are particularly relevant in this population, while maintaining a cautious and transparent interpretation consistent with the study design.
2.Mini Nutritional Assessment–Short Form (MNA-SF) was also used to assess nutritional status. I believe that tools more specific to hemodialysis patients, such as one of the PEW criteria, simplified PEW criteria, and MIS, should be used to assess malnutrition.
Response 2
We thank the reviewer for this suggestion and agree that dialysis-specific tools such as protein–energy wasting (PEW) criteria, simplified PEW definitions, or the Malnutrition–Inflammation Score (MIS) provide a more detailed characterization of nutritional status in patients undergoing hemodialysis.
However, the aim of the present study was not to diagnose malnutrition or to characterize PEW, but rather to explore the relationship between fatigue, sleep disturbances, and nutritional status from a clinical and functional perspective in a large, multicenter cohort of hemodialysis patients. In this context, the MNA-SF was used as a nutritional screening tool, widely validated and commonly applied in clinical practice, allowing early identification of patients at nutritional risk while incorporating functional and intake-related dimensions that may be related to fatigue perception.
As now explicitly stated in the Limitations section, dialysis-specific malnutrition scores were not incorporated due to the multicenter design of the study and the lack of homogeneous availability of the variables required for their calculation across all participating centers. Therefore, a pragmatic approach was adopted, prioritizing data comparability and sample size, without the intention of replacing more specific diagnostic tools.
We believe that this approach is consistent with the study design and objectives and allows a cautious interpretation of the findings, while also acknowledging the need for future studies incorporating dialysis-specific nutritional tools to further elucidate the underlying mechanisms.
Minor points
- In the Introduction section, the points should be more concisely described with enough information in 3 or 4 paragraphs; background, known, and unknown. The prevalence of fatigue and PEW should be commented on in the Introduction section.
Response minor 1
Thank you for this suggestion. The Introduction has been shortened and streamlined, preserving all references. In addition, an explicit mention of the high prevalence of protein–energy wasting (PEW) in hemodialysis patients has been added within the MICS framework. Malnutrition is addressed from a nutritional risk perspective, consistent with the aims and design of the study.
2. Please define the residual kidney function.
Response minor 2
Thank you for this comment. Residual kidney function has now been explicitly defined in the Methods section, described as the remaining renal capacity to maintain urine production and contribute to solute and excess fluid elimination, assessed by the presence of patient-reported residual urine output (yes/no).
3. In Figure 2, trend analysis (Jonckheere-Terpstra trend test) should be performed.
Response minor 3
We thank the reviewer for this suggestion. We have now performed a Jonckheere–Terpstra ordered trend test to assess whether fatigue and sleep disturbance scores increase across ordered categories of nutritional status (normal nutritional status, at risk of malnutrition, and malnutrition). A significant increasing trend was observed for both fatigue (p < 0.001) and sleep disturbance (p < 0.001). These results have been added to the Results section and are supported by the revised boxplots.
Note to the Reviewers
We would also like to inform the reviewers that two additional authors, Soraya Escribano-Loma and Cristina Cubas Sánchez-Beato, have been included in the manuscript in recognition of their substantial contribution to the critical revision of the manuscript and to the improvement of its scientific content, in accordance with the journal’s authorship criteria.

Round 2
Reviewer 1 Report
Comments and Suggestions for Authors
I thank the authors for their thorough response and the substantial effort made to reanalyze the data by treating fatigue and sleep disorders as independent domains. The removal of the composite variable and the clarification regarding the adapted nature of the PROMIS items adequately address the main construct validity concern raised in the first review round. Overall, the manuscript has improved considerably in both methodological rigor and conceptual clarity.
I have one final methodological point that should be addressed to ensure the technical quality of the manuscript prior to acceptance:
Internal consistency of the ad hoc subscales (remaining critical point).
Although a full psychometric validation is not required, summing four items to create a “Fatigue” score and three items to create a “Sleep Disorders” score effectively results in two ad hoc psychometric scales. Standard practice requires reporting internal consistency to justify item aggregation. At present, it cannot be assumed that the items are sufficiently correlated.
I therefore request that the authors calculate and report Cronbach’s alpha (or McDonald’s omega) for both the Fatigue (4 items) and Sleep Disorders (3 items) subscales in the Methods or Results section. Values >0.70 would provide reassurance regarding measurement reliability. This is a straightforward analysis that does not alter the study design but is essential to support the main variables.
Regarding the Discussion, the revised section on residual renal function (RRF) is appropriate and well developed. I appreciate the transparent reporting that the association with fatigue became marginal (p = 0.059) after variable separation, while the association with sleep disorders remained significant (p = 0.002). The added mechanistic explanation linking sleep impairment to fatigue in patients without RRF is convincing and strengthens the manuscript.
Minor points have been adequately addressed: Table 4 now clearly separates correlations for fatigue and sleep, and the inclusion of the additional co-authors is appropriately justified.The manuscript is now methodologically sound after removing the unvalidated composite variable. The only remaining minor requirement is the inclusion of internal consistency coefficients for the two subscales. Once this information is added, the article will be ready for acceptance.
Author Response
Review Report Form 1
Comments and Suggestions for Authors
I thank the authors for their thorough response and the substantial effort made to reanalyze the data by treating fatigue and sleep disorders as independent domains. The removal of the composite variable and the clarification regarding the adapted nature of the PROMIS items adequately address the main construct validity concern raised in the first review round. Overall, the manuscript has improved considerably in both methodological rigor and conceptual clarity.
I have one final methodological point that should be addressed to ensure the technical quality of the manuscript prior to acceptance:
Internal consistency of the ad hoc subscales (remaining critical point).
Although a full psychometric validation is not required, summing four items to create a “Fatigue” score and three items to create a “Sleep Disorders” score effectively results in two ad hoc psychometric scales. Standard practice requires reporting internal consistency to justify item aggregation. At present, it cannot be assumed that the items are sufficiently correlated.
I therefore request that the authors calculate and report Cronbach’s alpha (or McDonald’s omega) for both the Fatigue (4 items) and Sleep Disorders (3 items) subscales in the Methods or Results section. Values >0.70 would provide reassurance regarding measurement reliability. This is a straightforward analysis that does not alter the study design but is essential to support the main variables.
Regarding the Discussion, the revised section on residual renal function (RRF) is appropriate and well developed. I appreciate the transparent reporting that the association with fatigue became marginal (p = 0.059) after variable separation, while the association with sleep disorders remained significant (p = 0.002). The added mechanistic explanation linking sleep impairment to fatigue in patients without RRF is convincing and strengthens the manuscript.
Minor points have been adequately addressed: Table 4 now clearly separates correlations for fatigue and sleep, and the inclusion of the additional co-authors is appropriately justified.The manuscript is now methodologically sound after removing the unvalidated composite variable. The only remaining minor requirement is the inclusion of internal consistency coefficients for the two subscales. Once this information is added, the article will be ready for acceptance.
Respose 1:
As requested, we have now assessed the internal consistency of the two ad hoc subscales. Cronbach’s alpha was calculated for the fatigue subscale (4 items) and the sleep disturbances subscale (3 items), both showing acceptable internal consistency (α = 0.80 and α = 0.78, respectively). These results have been added to the Methods/Results section to support item aggregation.

Reviewer 3 Report
Comments and Suggestions for Authors
I am satisfied your revision. Now, I would like to recommend your manuscript be accepted for Nutrients. I am so happy to help your manuscript much better.
Author Response
Review Report Form 3
I am satisfied your revision. Now, I would like to recommend your manuscript be accepted for Nutrients. I am so happy to help your manuscript much better.
Respose:
We thank the Editor for the positive assessment of the revised manuscript. We appreciate the constructive feedback and the opportunity to further improve the clarity and quality of the work.
